# Big Data ETL Process and Its Impact on Text Mining Analysis for Employees' Reviews

Laura Gabriela Tanasescu, Andreea Vines, Ana Ramona Bologa * and Claudia Antal Vaida

Department of Computer Science and Cybernetics, Bucharest University of Economic Studies, 010374 Bucharest, Romania; laura.tanasescu@csie.ase.ro (L.G.T.); andreea.vines@csie.ase.ro (A.V.); antalvaidaclaudia12@stud.ase.ro (C.A.V.)
* Correspondence: ramona.bologa@ie.ase.ro

**Abstract:** Big data analysis is challenging in the current context for enterprises that would like to apply these capabilities in the human resource sector. This paper will show how an organization can take advantage of the current or former employees' reviews that are provided on a constant basis on different sites, so that the management can adjust or change business decisions based on employees' wishes, dissatisfaction or needs. Considering the previously mentioned challenge on big data analysis, this research will first provide the best practice for the collection and transformation of the data proposed for analysis. The second part of this paper presents the extraction of two datasets containing employee reviews using data scraping techniques, the analysis of data by using text mining techniques to retrieve business insights and the comparison of the results for these algorithms. Experimental results with Naïve Bayes, Logistic Regression, K-Nearest Neighbor and Support Vector Machine for employee sentiment prediction showed much better performances for Logistic Regression. Three out of the four analyzed algorithms performed better for the second, triple-size dataset. The final aim of the paper is to provide an end-to-end solution with high performance and reduced costs.

**Keywords:** text mining; web scraping; big data; sentiment analysis; machine learning





## 1. Introduction

Big data represents a revolutionary step forward from traditional data analysis, being characterized by three main components—variety, velocity and volume [1].

Moreover, as an emerging field of research and a practical method, big data has started an extensive discussion over the past years. The technology of the Internet, which has advanced in a fast-growing manner, offers, nowadays, new data collection channels for different businesses and management research domains. A big data approach started to be a must-have implementation in terms of data retrieving, storing, loading, and analyzing in many organizations, especially for the human resource area, which impacts or contributes directly to the performance of the company [2]. In today's context, even if good data can lead to better decisions, the high volume of the data collected from multiple external and internal sources can exceed the capacity of storage and analysis tools. Thus, to exploit the advantages deriving from Big Data analysis and from the use of Big Data analytics, there is a need to understand how data can be managed to improve decisions. Furthermore, recent studies have noted that there is still a sizable number of companies that fail to capture value from their big data investments [3], and even some that argue that big data may hurt rather than help companies [4]. As a result, there is insufficient understanding about how organizations should approach their big data initiatives, and scarce empirical support to guide value creation from such investments [5].

Moreover, in the past, it has been observed that human resources is a domain that differentiates a lot from other areas where artificial intelligence has been applied. One of the challenges is represented by the complicity of human resources outcomes. A second

problem for data science is that many of the important outcomes in HR are relatively rare events, especially in small organizations. Additionally, since data science and machine learning require a large amount of data for analysis, the results obtained from applying different techniques can be poor [6].

Recently, in many artificial intelligence areas, text categorization has also gained importance, depending also on the size of text data and information. Due to the enormous potential of sentiment analysis for smart education and data-driven decisions, this technique has been deemed as a powerful analysis for identifying and classifying sentiments for multisource and multimodal data.

This research paper aims to propose an end-to-end solution for the companies that would like to understand and make data-driven decisions, based on employees' feedback. The research will not only focus on how the data should be retrieved in an optimal approach, but also how it should be stored and analyzed, so that it can finally produce conclusions that can be used by the business units.

To meet the purpose of this paper, the research methods consisted of data gathering through web scraping from various specialized websites, data cleaning and structuration to align data types and prepare the datasets for further analysis, data enrichments, aiming to label the observations and prepare the datasets for supervised machine learning algorithms, data analysis for a better understanding of the content, and analysis of the algorithms' results for an assessment of the best performing one for classification problems in the sentiment analysis area. For the later part, four classification algorithms were considered and applied on the two datasets obtained: Naïve Bayes, Logistic Regression, K-Nearest Neighbor and Support Vector Machine.

As a first step, which concerns the collection of data, according to a paper published by Kitchin and McArdle, scraped websites represent a big data source that generates a high volume of data [7]. Moreover, web scraping represents a technique used to extract data from different websites and transform it from an unstructured format to a structured standardized type by simulating human behavior. It is currently used in different domains such as online price comparison, website change detection, weather data monitoring or web research [8].

On the other hand, a very important area in which we generally use the big data collected through various methods to add value for our process is the field of artificial intelligence, which is constantly growing and solves different real-world problems. One of the artificial intelligence research areas is natural language processing, which attempts to process and classify human language data. Moreover, one of the most important uses of natural language processing is sentiment analysis, which is a technique that can help us build systems that try to identify and extract options or real feelings from oral speaking or even written texts. This type of analysis is very important for every organization, as it can take into consideration employee or customer opinions, and accordingly the organization can improve the products or their business [9].

This paper aims to present a complete solution in order to analyze current and former employees' reviews, which are provided on the Indeed website for a particular company, and use the details received for future reviews classification and also for empowering internal data-based decisions and process optimization for one organization. Section 2 presents the background and related work for the algorithms and methods that are used—starting with web scraping for data extraction and sentiment analysis techniques for data analysis. For each algorithm, there is a brief introduction and a description of the used methodology. Furthermore, Section 3 describes the methods and results obtained by using a dataset with Amazon reviews (previously collected using web scraping techniques) and presents all the details starting with the dataset presentation, details regarding the extraction process using web scraping techniques and the text mining methods used for sentiment analysis. Section 4 offers the conclusions of this research, possible improvements that can be taken into consideration for future work in the same area, as well as opportunities for extending the current research.

## 2. Background and Related Work

### 2.1. Web Scraping

Web scraping can also be defined as a process that involves extracting and combining contents from the Internet in a systematic way, where a software agent mimics the human experience of Web browsing interaction [10]. From an operational point of view, it can be considered a manual activity, which implies copying and pasting the information. The difference here is that this task is performed in an automatic way using a virtual computer agent [11]. It might be also considered an extraction process that transforms unstructured information from the Internet into a structured format that can be easily read and used in different analyses.

The web scraper accesses the web site using a request, finds specific elements using the HTML code, extracts and saves them into a structured format using a data frame.

This technique of extracting unstructured data from the web sites can be applied in different scopes. For example, it can be used to compare product prices across multiple websites, to extract ads from linked pages or to cover incomplete data issues from statistics or to complete official datasets with more information [8,12]. Regarding the human resources area, web scraping can also be used to find available jobs on different websites and to classify them using Naïve Bayes algorithms.

According to Hillen, there are several advantages for web scraping. One first advantage is the low costs, as it can be free depending on the source program that is used. It can also be used anytime, depending on how frequently data need to be refreshed (hourly, daily or monthly). Another advantage might be that it can be customized according to the needs of the data analyst—compared to different datasets already existing that have predefined columns and data types [12].

Web scraping might be the easiest solution to extract datasets from the Internet, but it also comes with some challenges. One of them is that the web scraper is conditioned by the number of records available on a single page. For example, there are websites with only ten entries per page and it makes it difficult for the web scraper to go through all pages in order to extract data. We can also mention websites configured with infinite scrolling without reaching the end of the page [13]. Another limitation would be regarding the volume of the data and how long it will take a software program to extract all data if it has to extract all records each time or only new/delta entries. Data quality is another aspect that affects the web scraping process. In order to keep the accuracy of data, there should be different validation rules included in order to make sure that all the information is consistent and can be used in further analyses.

Web scrapers can be implemented using several approaches, depending on the requirements received. They can be divided into two main categories—manual and automated web scraping. The manual category represents extracting data from the Internet by copying the content and pasting it into a structured format. It is the simplest form of web scraping, but it might take longer depending on the size of the dataset that needs to be retrieved. There are multiple automated web scraping techniques, such as HTTP programming, HTML parsing or DOM parsing; all of these involve navigating through the website structures and extracting the information. Different web scraping software tools can also be used in order to avoid the need to write the program and test it. Another interesting approach is using computer vision web-page analyzers that identify different blocks of content of the web page and extracts the information that is needed [11,14].

Several libraries can be implemented in Python in order to extract data. BeautifulSoup is the easiest to use and implement, but it is not available to be used for JavaScript websites. For those websites, Selenium should be used, which is slower, as it imitates human behavior by navigating through the pages and extracting the information. Another popular Python library is Scrapy, which is actually a framework, developed for web scraping, as it is faster than the previous two mentioned, but it cannot handle JavaScript websites [13,15]. All libraries mentioned have different advantages depending on the scope of the project where web scraping is required. Their main disadvantage is that they cannot be used individually

for a complete process, so at least two libraries need to be combined to retrieve all the information needed from the Internet.

*2.2. Artificial Intelligence and Sentiment Analysis*

Sentiment analysis is seen to be, in previous research realized in this domain, a very restricted NLP problem. For this kind of analysis, we need to be able to conclude if there are any positive or negative sentiments for each target sentence or phrase. However, despite these facts, the methods that were discovered in the research field are working correctly in terms of Information Retrieval. The main struggles identified at the moment are pointing on how to handle in the best way negations or names of entity recognition, as well as the challenges of being able to deal with irony or sarcasm [16,17]. Other studies have proved that NLP needs to deal with different levels of analysis. Depending on the form existing for the target subject (either text, document, or maybe linked sentences), different NLP and types of sentiment analysis can be applied. Therefore, it is very important to distinguish the levels of the analysis that will determine the tasks of the analysis: document level, sentence level or entity level [18].

Many papers in this domain are therefore following the general strategies. For example, the papers detailed by Vechtomova and Karamuftuoglu [19] proposed the use of lexical cohesion, hence the use of physical distance between the collocations in order to realize a rank of the documents. At the same time, Vechtomova further proposes one more method measuring the distance between subjective words [20]. Additionally, there are some other examples that apply the known supervised methods of Artificial Neural Networks and Support Vector Machines in order to classify sentiments [21].

Sharma et al. have also conducted a study on sentiment analysis for big data. They have worked on demonstrating an overview of the current updates in opinion mining. They discovered that sentiment analysis has become very popular in this research field, and they also stated that a lot of research has already been performed, even though we still have challenges pointing to unstructured data. Moreover, they have declared that the most repeated lexicon source is wordnet, and that supervised techniques provide more accurate results than the dictionary techniques [22]. Dandannavar and Mangalwede have also presented a survey on some other techniques of sentiment analysis using textual data. They have presented in their study advantages and disadvantages of each method in a summarized way, and the conclusion was that, in order to be able to perform sentiment analysis, multiple techniques could be used, some methods being based on lexicon, some using training sets, and some using both of them. The analyzed methods are specific to a domain, and most of them study English and Chinese languages. Therefore, very few studies have been conducted on sentiment classification for other languages [23].

Finally, Hu and Bing [24] also presented an important method of the dictionary-based approach. In this one, a set of opinion words are manually collected, and this set is grown by searching in different areas (such as the well-known corpora WordNet or thesaurus), so that other synonyms and possible other antonyms can be found. The new words are then added to the seed list and the next iteration is being triggered. All this process stops in the moment when no other words are found. However, this dictionary-based method has major disadvantages, such as the inability to distinguish words that express opinions from those that are related to a specific context. Qiu and He used the exact same approach to identify sentiment sentences in the advertising domain. Their proposal in this new context was to have a syntactic parsing, as well as a sentiment dictionary, but also a proposed rule for topic word extraction, which finally showed effectiveness in the process of extracting and selection the advertising keywords [2,25].

For the general classification of sentiments, we can use two of the most popular approaches: subjective lexicon and machine learning.

The first approach consists of a collection of words where each word has a score indicating the positive, negative, or neutral nature of the text. In this approach, the sentiment of the given text is represented by polarity and subjectivity. For a given piece of

text, aggregation of scores of subjective words is performed, so positive, negative, neutral, and objective word scores are summed up separately. In the end, four scores will be obtained. The highest score will give the overall polarity of the text. The polarity measures how positive or negative a text is on a scale of $(-1,0,1)$ and subjectivity represents how much of an opinion it is vs. how much of a fact [26].

On the other hand, the machine learning technique is an automatic one. Classifications will be performed using the text features. Therefore, in this case, we can also talk about supervised and unsupervised learning. For the supervised one, the system will be trained using labeled training examples [27]. Each class will represent different features and will have a label associated with it. When a word arrives, the features are compared and labeled with that class with which it has the maximum matching [26]. Additionally, unsupervised learning requires the deduction of functions for presenting possible unknown structures for the unlabeled data. This technique does not need a supervisor, so it means that the system must have the ability to learn independently with training based on unlabeled data received as input [28].

## 3. Methods and Results

In order for our research to provide a clear and correct data mining process, we will implement the CRISP-DM (Cross Industry Standard Process for Data Mining) methodology, which helps us by defining a process model that can provide a framework for carrying out data mining projects. In the following part, we will observe that the life cycle of the data mining project is broken into six parts [29].

### 3.1. Business Understanding

This research objective is to observe real feedback from employees and former employees, so that a company can improve itself based on this feedback. Moreover, it is important to underline the fact that we can get valuable information from publicly available reviews; therefore, there is no need to collect any additional data, especially from former employees, a fact that can be pretty difficult. Last, but not least, we also want to propose a process that requires a minimum of manual work, so that we can again reduce costs and time in this objective of finding out people's feelings towards an organization.

### 3.2. Data Collection

For our next analysis, data were collected automatically from different job-related sites such as Indeed (https://www.indeed.com/cmp/Amazon.com/reviews (accessed on 4 January 2022)) or Glassdoor (https://www.glassdoor.com/Reviews/Amazon-Reviews-E6036.htm (accessed on 4 January 2022)) to retrieve as many reviews as possible regarding a company. The objectives were to realize what is the overall employee or former employee sentiment, what are the best words to explain their sentiments and also to build a prediction algorithm for the newest reviews that are coming on the concerned platforms and automate this process as much as possible.

#### 3.2.1. Web Scraping Process

The web scraping process that extracts data as reviews from different websites can be observed below in Figure 1. It was developed using the BeautifulSoup library to get the information and parse the HTML code and request library to send HTTP calls for the website to retrieve specific data.

The first two steps describe the website and the data that are needed for the web scraping process. The extracted source entities (such as names, descriptions, and links) are stored in the web page structure, so they will be retrieved from the code source using HTML DOM or XPath methods.

Those identifiers will be stored in a metadata table to be parsed as parameters. This will allow for using the same code source for different websites even if they have different identifiers for a specific element—for example, the title of the review is extracted from the

h2 tags that have the itemprop named "author" for the Indeed.com website. On the other hand, the rating is available for all the reviews found on the Glassdoor website using the span selector with the class called "authorInfo". The structure of the table can be analyzed below (see Table 1). This table allows one to use parameters in order to define the company and the website for which the user wants to extract data—in this, the source code will not be modified.

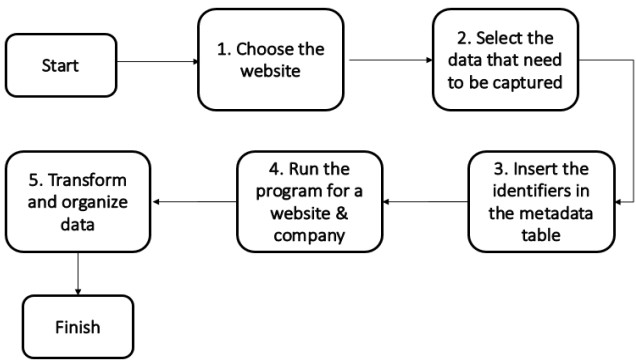

**Figure 1.** Process of web scraping (source: authors own study).

**Table 1.** Metadata table.

| Website | URL | Element | Selector | Type | Value |
|---|---|---|---|---|---|
| Indeed.com | https://www.indeed.com/cmp/Amazon.com/reviews | author | h2 | itemprop | author |
| Glassdoor | https://www.glassdoor.com/Reviews/Amazon-Reviews-E6036.htm | author | span | class | authorInfo |

In Figure 2, there can be seen the structure of a review that corresponds to the function used to extract the elements—the review, title, and feedback. The author is a string that contains multiple elements, such as job function, job status (if it is a current employee or not), location, and the review data.

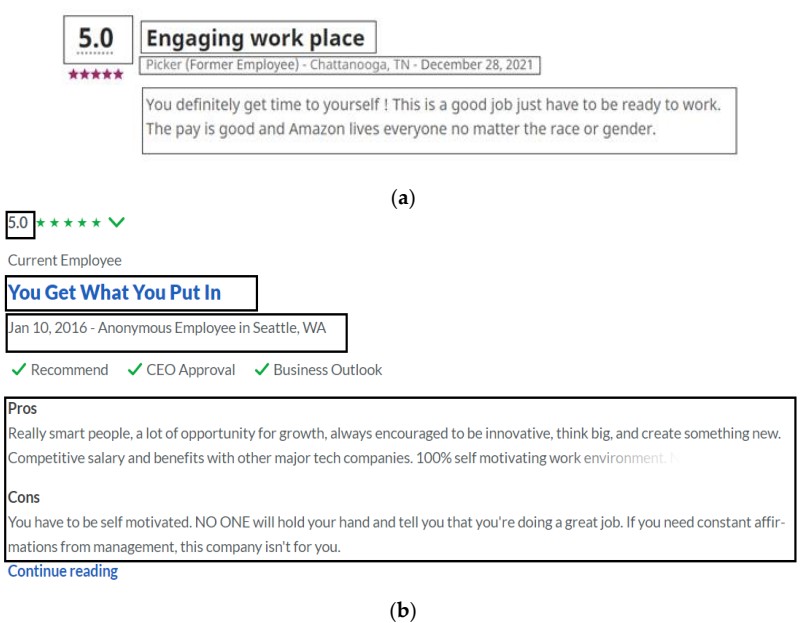

**Figure 2.** Structure of a review: (**a**) Indeed.com; (**b**) Glassdoor.

The fourth step is to create the source code that extracts data from the website. It connects to the URL added as a parameter to automate the process to be used for multiple

companies and creates a request to access data. If the connection is successful, a list is created, and it navigates through all pages to extract the reviews.

The last step of the web scraping process is the transformation of data into a structured format that can be used for further analysis. The title column was split into multiple columns to retrieve the job name, location, employee status, and the data; all columns were converted into appropriate formats to be further used in the analysis.

The approach of using metadata and parameters improves the scalability of the process as there can be used the same code script to extract data from multiple websites or to use the same extract but retrieve data for multiple companies. However, the disadvantage of this approach is that the process is slower when we are using parameters.

In Table 2, there can be observed the timings obtained while running the web scraping script using parameters instead and the values using Databricks. This is a resource provided by Microsoft Azure and it is used in order to process large workloads of data by running queries in different languages, such as Python, R, Scala or SQL. For the compute power, it uses a cluster with 14 GB memory and 4 cores, so there will be no need for additional infrastructure to process data.

**Table 2.** Timing for web scraping.

| Method | Time | Number of Records | Cluster Type |
|---|---|---|---|
| Using metadata | 3.40 h | 66,854 | Runtime version: 10.3 |
| Without metadata | 1.58 h | 67,141 | Worker type: 14 GB Memory, 4 cores<br>Driver type: 14 GB Memory, 4 cores |

Despite the timing that might be different while using metadata, if multiple sources should be included, the transformations should be performed separately as they might be different from one data source to another. This will require more effort from the development point of view in order to create different transformations—such as splitting columns, concatenating feedback columns if they are stored separately, data conversions, and so on.

Moreover, the complete web scraping can be automated using an orchestrator in order to have a scheduled job to retrieve all new reviews that are posted on the websites.

### 3.2.2. Data Analysis and Distribution

The web scraper presented above is used to retrieve all employees' reviews posted on Indeed.com—the first job site in the world with over 250 million unique visitors every month. The final dataset contains reviews for Amazon, added by different employees worldwide. The time range for all entries extracted is from 15 February 2012 until 3 January 2022.

In the Table 3 below, there can be observed the overall rating of 3.49 across all years. The lowest rating was recorded for management (2.92 out of 5.00) and the greatest for pay benefits (3.47 out of 5.00) close to the average overall rating—which means that most of the employees are satisfied with their salary packages.

**Table 3.** Averages of the important variables obtained from collected data.

| Year | No of Records | Avg Overall Rating | Avg Work-Life Balance | Avg Pay Benefits | Avg Job Security | Avg Management | Avg Culture |
|---|---|---|---|---|---|---|---|
| 2012 | 539 | 3.71 | 2.99 | 3.38 | 2.92 | 2.98 | 3.19 |
| 2013 | 1265 | 3.78 | 2.99 | 3.32 | 2.85 | 3.01 | 3.24 |
| 2014 | 2032 | 3.74 | 3.05 | 3.33 | 2.9 | 2.91 | 3.15 |
| 2015 | 2981 | 3.67 | 3.05 | 3.41 | 2.94 | 2.93 | 3.15 |
| 2016 | 4224 | 3.67 | 3.1 | 3.52 | 3.04 | 2.97 | 3.21 |
| 2017 | 10,168 | 3.67 | 3.2 | 3.61 | 3.14 | 3.12 | 3.31 |
| 2018 | 13,552 | 3.55 | 3.17 | 3.55 | 3.03 | 3.07 | 3.24 |
| 2019 | 16,820 | 3.52 | 3.07 | 3.46 | 2.93 | 2.91 | 3.12 |
| 2020 | 15,187 | 3.47 | 3.17 | 3.46 | 2.98 | 2.91 | 3.15 |
| 2021 | 10,688 | 3.05 | 2.88 | 3.32 | 2.71 | 2.57 | 2.76 |
| 2022 | 45 | 3.38 | 2.98 | 3.42 | 2.89 | 2.82 | 2.78 |
| Overall | 77,501 | 3.49 | 3.1 | 3.47 | 2.96 | 2.92 | 3.13 |

In the graph below from Figure 3, there can be seen the top five locations with the greatest number of reviews—where the employees from Phoenix, AZ added 1091 reviews. The next location by the number of records is Seattle, WA with 1078 entries, followed by San Bernardino, CA (875), and Baltimore, MD (848).

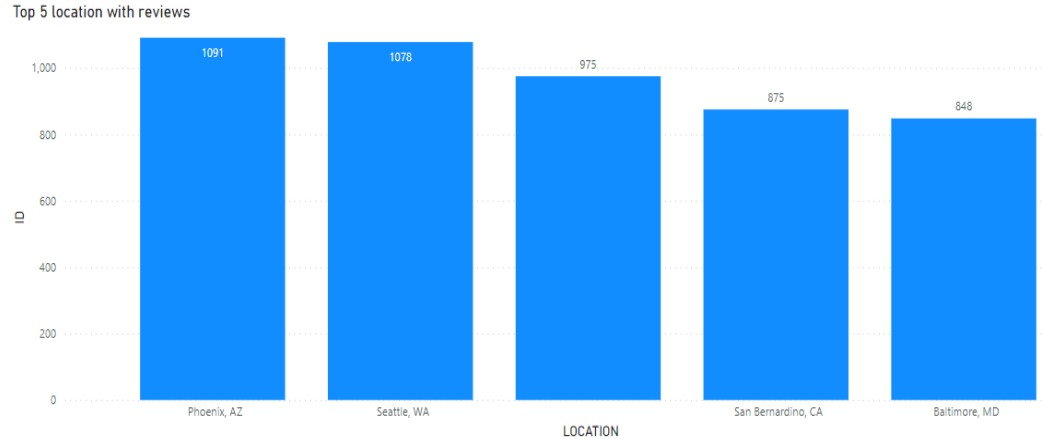

**Figure 3.** Top five locations chart for the provided reviews.

A similar bar chart grape from Figure 4 to display the job positions with most of the reviews. We can notice that most of the reviews come from warehouse employees (Warehouse Associate, Fulfillment Associate, Packer, Picker or Stower) and there are fewer reviews posted by IT employees (where the job position contains the 'Software' keyword). Moreover, we can see the positions with the highest numbers of reviews described in Figure 5.

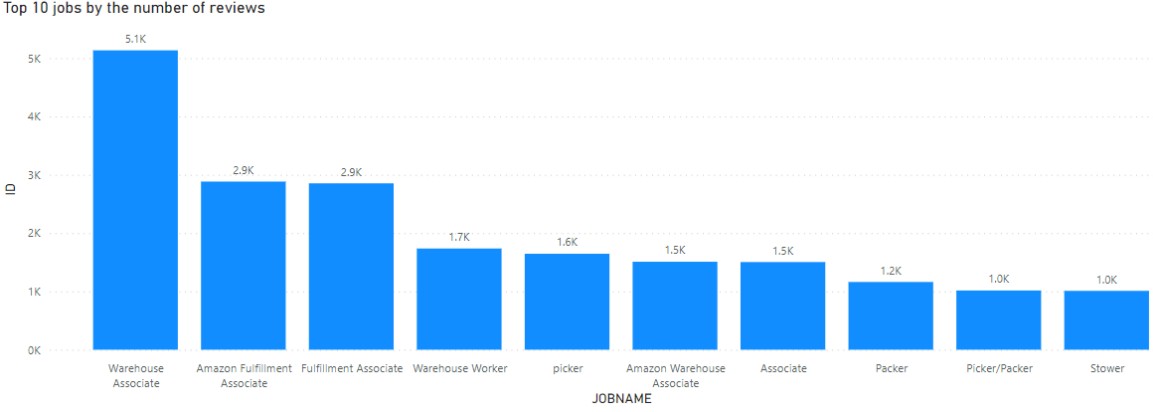

**Figure 4.** Top 10 jobs distributed based on number of reviews.

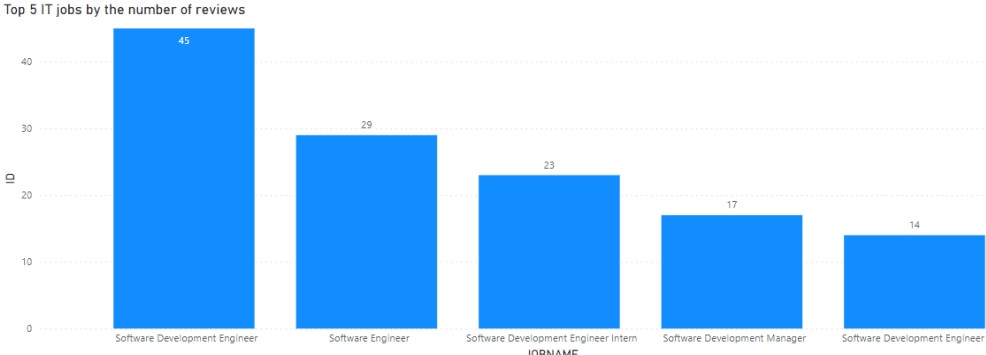

**Figure 5.** Top 5 jobs distributed based on number of reviews.

One more way to visualize the entire dataset and different metrics associated with it is to also import the entire dataset into a business intelligence tool and observe the distribution of data per different categories.

For example, in Figure 6, one can observe the rating distribution by the number of records using bar-chart graphs. In the first histogram, the distribution of job security is represented, and it can be noticed that most of the respondents consider that Amazon offers 3 out 5 for job security. Moreover, as it can be observed in the fourth graph, almost 10k of the reviews from the dataset seem to be satisfied with the pay benefits received, as they offered the maximum value for the rating. Nonetheless, the visualizations have been realized using Power BI for a more practical way of understanding the data.

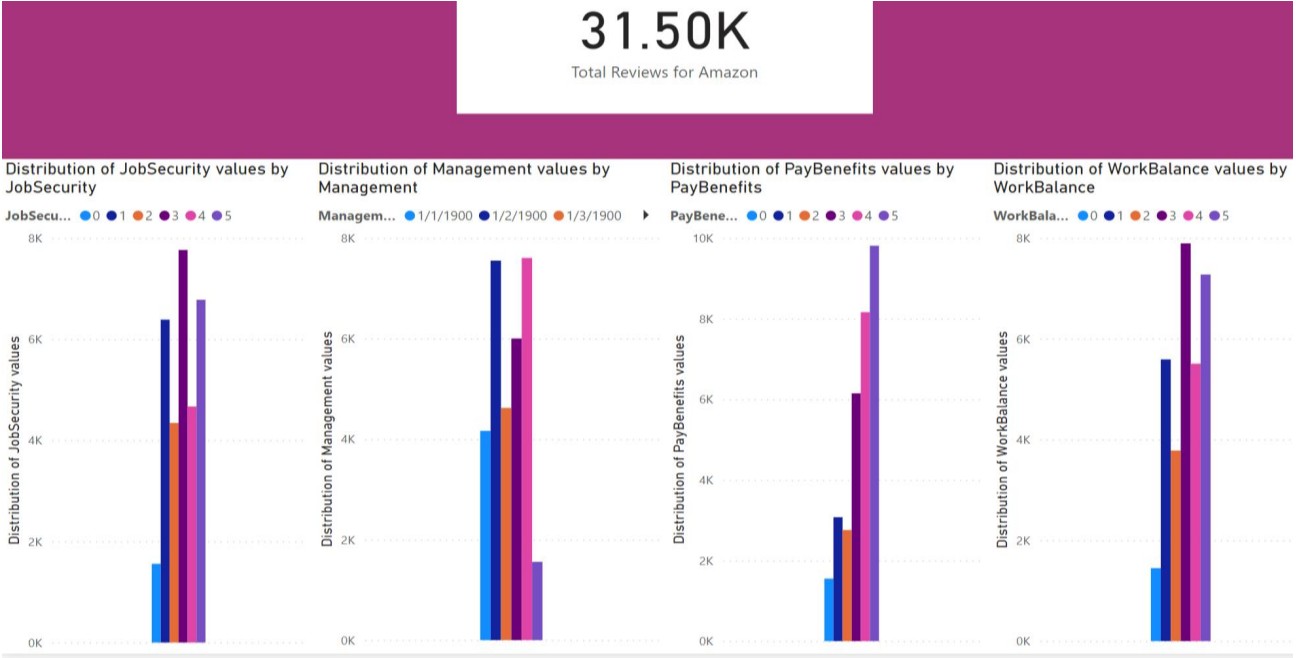

**Figure 6.** General distribution observed on the input data.

### 3.3. Data Preparation

Considering the initial dataset retrieved from the web scraping step, we now have different information about the opinions that former or actual employees of the company have provided on the sites previously mentioned. A subset of the current dataset can be observed in Figure 7.

In order to continue with our data analysis, we will first make sure that we are following the steps mentioned before so that we can have conclusive and correct results.

First of all, after collecting the data that was previously mentioned, we will perform some steps in terms of cleaning the current data. Apart from the following section that presents the most common ways of data pre-processing, we also took care of the missing values. First of all, we did this by dropping the review if the details of the person leaving it were also missing. Second of all, if just the review was empty, we assumed that the reviewer did not have anything else to say about the organization; therefore, this has also been taken care of in the feature engineering section, where the bag of words method can represent this even if the string is 0.

In addition, specifically for our text data, we will introduce the associated steps for the pre-processing of text data:

a. Tokenization consists of breaking complex sentences into words, and it will understand the importance of each word as well with respect to the sentence.

b. Stemming is the process of normalizing words into their base form or root form.

    c.      Lemmatization will group together different inflected forms of the word. This is similar to stemming, because it will map several words having the same meaning of the root output in a proper word.

    d.      Removing special characters.

    e.      Removing extra new lines.

    f.      Removing accented characters.

    g.      Removing contractions will mean the contribution to text standardization and expand the words or combinations of words that were previously shortened by dropping letters and replacing them with an apostrophe.

    h.      Removing stop words is the process that will remove the unnecessary words of the sentences, in order to reduce the size of the corpus.

| | A | B | C | D | E | F | G | H | I | J | K | L | M |
|---|---|---|---|---|---|---|---|---|---|---|---|---|---|
| | Id | Title | JobName | JobStatus | Location | Date | Feedback | OverallRatin | WorkBalanc | PayBenefits | JobSecurity | Manageme | Culture |
| | 1 | Okay place | Shift Manaç | Former Em¡ | Lansing, MI | 2021-12-17 | The only thi | 3 | 3 | 4 | 3 | 3 | 3 |
| | 2 | Good place | Software D | Current Em | Seattle, WA | 2021-12-18 | Amazon is ¿ | 5 | 4 | 5 | 3 | 4 | 3 |
| | 3 | Crazy ppl ar | Dock Work | Current Em | Romulus, M | 2021-12-18 | They promi: | 2 | 2 | 3 | 1 | 1 | 1 |
| | 4 | Instead, the | Warehouse | Former Em¡ | Georgia | 2021-12-18 | While work | 2 | 0 | 0 | 0 | 0 | 0 |
| | 5 | Love love lo | Warehouse | Former Em¡ | Colorado S¡ | 2021-12-17 | Loved the j¢ | 5 | 5 | 5 | 5 | 5 | 5 |
| | 6 | Fun place t¢ | Overnight S | Former Em¡ | Los Angeles | 2021-12-17 | Its a good p | 5 | 5 | 5 | 5 | 5 | 5 |
| | 7 | Physically h | Picker/Pack | Current Em | Easton, PA | 2021-12-17 | Can be bori | 4 | 4 | 5 | 4 | 5 | 4 |
| | 8 | Physically d | Amazon W¿ | Current Em | Campbellsv | 2021-12-17 | Physically d | 3 | 3 | 5 | 5 | 1 | 1 |
| | 9 | Good place | Warehouse | Current Em | Fort Wayne | 2021-12-17 | I've only be | 4 | 5 | 5 | 3 | 4 | 5 |
| | 10 | Decent wor | Process Gui | Current Em | Euclid, OH | 2021-12-17 | Managers c | 3 | 1 | 2 | 2 | 2 | 2 |
| | 11 | it's alright | Warehouse | Current Em | Breinigsville | 2021-12-17 | Decent pay | 4 | 5 | 4 | 3 | 4 | 4 |
| | 12 | It's a job fo | Order Picke | Former Em¡ | Salt Lake Ci | 2021-12-17 | They say sa | 1 | 1 | 1 | 1 | 1 | 2 |
| | 13 | Bad | Warehouse | Former Em¡ | Hebron, KY | 2021-12-17 | It's Amazon | 1 | 1 | 3 | 1 | 1 | 1 |
| | 14 | The associa | Amazon Ful | Current Em | Rialto, CA | 2021-12-17 | Lack of con | 1 | 1 | 4 | 1 | 1 | 1 |
| | 15 | Great bene | Process Ass | Former Em¡ | Chattanoog | 2021-12-17 | It is a decer | 3 | 3 | 3 | 2 | 2 | 2 |
| | 16 | It's OK. Ove | Transportat | Current Em | United Stat | 2021-12-17 | It's OK. Ove | 3 | 2 | 3 | 3 | 3 | 3 |
| | 17 | Never enou | Warehouse | Former Em¡ | 5501 Holab | 2021-12-17 | I did learn c | 3 | 3 | 3 | 1 | 2 | 1 |
| | 18 | It's okay. | Picker | Former Em¡ | Charlotte, N | 2021-12-17 | Working at | 2 | 1 | 4 | 4 | 2 | 5 |

**Figure 7.** Overview of the final dataset.

    At this moment, since we have a cleaned dataset for text analysis, we will continue with the discussions and processing of the data according to the two methods that we will apply further, to achieve our end goal.

### 3.4. Data Modeling

3.4.1. Lexicon-Based Approach for Labeling Initial Set of Data

    After data were cleaned, we were having clearer words that were relevant for each review provided on the platforms mentioned before, but not the sentiment associated with each review. Therefore, based on classical methods that can provide us with a good text mining analysis, we can conclude that the current dataset is not appropriate for supervised machine learning techniques, since we do not have previous history on the review sentiment to learn from.

    Moreover, at this moment in our research, there is no need to consider splitting the dataset for any further processing of the data, since we will not be able to compare results with previous sentiment associated with the text.

    Considering the other approaches for our scenario, it is clear that a lexicon-based approach will help us improve the original dataset with a sentiment for each review provided. For our case, we have chosen TextBlob, which is a Python library used for natural language processing, to achieve these tasks. This library also gives easy access to different

lexical resources and allows the user to work with classification, categorization, and some other tasks.

For lexicon-based approaches, the sentiment will be defined by its semantic orientation and by the intensity of each work in the sentence. This also requests a predefined dictionary classifying negative and positive works. In general, a text will be represented by a bag of words and, after the assignment of individual scores to all the words, a final sentiment is calculated [17].

TextBlob will return the polarity and the subjectivity of a sentence. Polarity, which lies between −1 and 1, can mean at −1 a negative sentiment, and at 1 a positive one. Negation works reverse the polarity. Subjectivity lies between [0, 1]. Subjectivity quantifies the amount of individual opinion and factual information contained in the text. The higher subjectivity means that the text contains personal opinion rather than factual information. TextBlob has one more parameter—intensity. TextBlob calculates subjectivity by looking at the 'intensity'. Intensity determines if a word modifies the next word. For English, adverbs are used as modifiers ('very good') [30].

However, before we start with the application of the lexicon approach on our entire dataset, we would like to test this method's performance as well. Since we have previously expressed that, initially, this is not a pre-labelled dataset, we have decided that, in order to test its performance, we can manually label some of the inputs, apply the lexicon method and compare real sentiment with predicted sentiment.

Therefore, we have taken a subset from the original dataset and added the associated sentiment. We have performed the same methodology expressed before in order to prepare the data, and afterwards we have applied TextBlob to predict the sentiments. Once this step was finished, we compared the predicted values with the real ones, and the results from this step can be seen in Table 4.

**Table 4.** Summary of results for testing texblob method.

| Algorithm | TN | FP | FN | TP | Accuracy | Precision | Recall | F1-score |
|---|---|---|---|---|---|---|---|---|
| TextBlob lexion method | 71 | 25 | 35 | 77 | 71% | 75% | 69% | 72% |

We observe that the performance indicators obtained are very close to the ones retrieved from similar tests using the lexicon method, as it is mentioned in related research [31]. Moreover, in order to be able to feed to the next algorithms more data, we will use this to actually label all our records needed for training using the above approach, so that we can train our machine learning algorithms with more data than what we could manually label.

Therefore, since we decided to continue with this method to label all our reviews, we applied again all the pre-processing steps on the initial dataset and used TextBlob for all the records. To be able to observe more visualizations on our data, we have imported the newly created dataset into a business intelligence platform.

Therefore, we can also see now, with the enhanced dataset, the distribution of the predicted sentiments in Figure 8, using the visualizations realized in Microsoft Power BI more frequently:

As a conclusion on the sentiments predicted, we can see that we have a majority of positive reviews, former employees tend to provide even more feedback than the current ones, and the high rates in terms of sentiment polarity come from the persons that have also high scores for pay benefits and see a work life balance.

Now, after applying the lexicon-based approach, not only do we have multiple pieces of feedback from different people regarding the company, but we also have the associated sentiment, built on the words that have been discovered in the review.

Finally, even though we do not have the best performance, so we cannot be 100% of all the predicted sentiments that were added as certain sentiments to our dataset, we consider that, for the moment, this was the only solution to proceed further with such a high amount

of data. Of course, in time, we can also manually label even more data and try to improve the accuracy.

**Figure 8.** Dashboard of sentiment distribution obtained from enhanced dataset.

3.4.2. Machine Learning Techniques and Data Processing for Sentiment Classification

The next step for our analysis is to continue the text mining analysis; however, our next approach will be focused on supervised machine learning techniques, which will require a labeled dataset.

Considering that we now have a variable "sentiment", which is already known for each review, we can now train a part of the initial dataset using supervised algorithms in order to create different classification models.

The end goal is to test the most performant algorithms used for this kind of analysis and understand which is the one that provides the best performance. Finally, this one will be used for prediction in our integrated solution, so that we can predict new reviews every time we collect more data.

Before proceeding to the training step itself, we will also take care of the feature extraction associated with the text data. For our case, we will use Bag of Words, which is a natural language processing technique for text modeling.

To explain the approach, we can say that a bag of words is actually a representation of the text data, which will describe the occurrence of the words within a document. While text data can be messy and unstructured, we know that machine learning algorithms always prefer structured and well-defined input. Therefore, by using Bag of Words, we can eventually convert the text that may vary in length, to a fixed length vector. Moreover, as machine learning works with numerical data, we will also convert, by using this technique, text data into its equivalent vector of numbers.

Finally, for this specific task of transforming the data in the needed format, we will continue for our data to use the Bag of Words model using sklearn in Python, by introducing the function CountVectorizer().

So, considering the fact that text data have been cleaned and pre-processed even before applying the lexicon approach, after the above feature extraction added specifically for

the machine learning part, we can see that we have good training data for the moment to proceed with our training and testing [32].

However, for this second part of machine learning analysis, we would like to extend the training and testing dataset. Therefore, we will run two different tests with the same algorithms: the first one will be the previously presented one in the main data discovery section (around 31,000 reviews collected from Indeed with the sentiment attached to the review, obtained by following the steps presented before) and a second dataset with around 90,000 reviews, which contains data from Indeed and Glassdoor. We can observe that we added more data to be consumed and two sources from where we collected this.

In addition, all the steps regarding our data preparation have been followed for both datasets.

Therefore, we are introducing the new dataset in order to observe how the chosen algorithms will perform when we have more input data for training, as well as when these data are being taken from more data sources.

In order to decide at this time what algorithms to implement for the next phase, we have looked to similar studies of text classification, to observe which are the most popular ones that also perform the best and do not require too much computation power and resources, since we want to also minimize these for now. So, as Thangaraj and Sivakami mentioned [33], Logistic regression and Naïve Bayes are the most widely used parametric classification algorithms, while Support Vector Machine (SVM), Decision Tree, KNN and Neural Networks are their non-parametric counterparts. Since both Decision Tree and Neural Networks require more resources and more time, especially on a large dataset, we have decided for now to proceed with the other popular ones.

The chosen algorithms for this next step are some popular ones used in this scenario by other researchers such as Naïve Bayes, Logistic Regression, KNN and Support Vector Machine, as they are described below, considering the main advantages they benefit from:

- Naïve Bayes— Naïve Bayes classifiers are a collection of classification algorithms based on Bayes' Theorem. It is not a single algorithm but a family of algorithms, where all of them share a common principle, i.e., every pair of features being classified is independent of each other. Bayes' Theorem finds the probability of an event occurring given the probability of another event that has already occurred [34]. In addition, Naïve Bayes is mainly used for data pre-processing applications due to the fact that it does not require many resources. Bayesian reasoning and probability inference are all involved in predicting the target class, while attributes play an important role in the classification. Therefore, it might improve the overall performance of the assignment of different weight values to attributes [33].

- Logistic Regression—Logistic Regression is a type of linear regression that does not require linearity between the independent variables, while the dependent ones do not require a normal distribution. This method can be applied regardless of the types of variables: discrete, continuous or binary. This algorithm proved to perform for classification problems but had similar results to others that are easier to apply [35]. This algorithm, working in supervised learning, selects the best subjects to be labeled to achieve a good classification, a fact that represents an opportunity to reduce temporal costs. Moreover, active learning is engaged to find the best subject possible to label these machine learning models, a fact that represents a growing field of research in the area of text mining [33].

- KNN—K-Nearest Neighbor is a nonparametric classifier that learns from the similarities between classes. It uses a distance function for the relevant features and once a new record is added to the model, it analyzes its pattern and compares it with the nearest neighbors, adding it to the most similar class [36]. Even though this algorithm is robust for noisy data, deciding the exact value of k can be a complicated subject. Computation complexity further increases, with an increase that can be seen in dimensionality as well. Therefore, to reduce the cost of computing the k value, tree-based KNN is generally used [33].

- SVM—Support Vector Machine is a supervised machine learning algorithm used for both classification and regression. The objective of the SVM algorithm is to find a hyperplane in an N-dimensional space that distinctly classifies the data points. The dimension of the hyperplane depends upon the number of features. Compared to newer algorithms such as neural networks, they have two main advantages: higher speed and better performance with a limited number of samples (in the thousands) [34]. SVMs are particularly suitable for high-dimensional data, while the complexity associated with the classifiers depends on the number of the support vectors instead of data dimensions, a fact that produces the same hyper plane for repeated training sets [33].

In the end, we aim to see a comparison between the performance on these four algorithms and between the tests run on different amounts of data, as well as the time taken for running them and the resources consumed.

Moreover, before proceeding to building the models, we will also split the data between train and test data. For this step, we will introduce a sklearn function called train_test_split and we will configure this to take as train data 60% of the total dataset. This split, with the same settings, is applied for both datasets.

*3.5. Evaluation*

In the following section, we will observe the performance indicators received from all the algorithms applied for training associated with the two sets of data that we have considered during the analysis. The end results can be seen with all the associated details in Table 5.

**Table 5.** Summary of results after initial training.

| Algorithm | Dataset | TN | FP | FN | TP | Accuracy | Precision | Recall | F1-score |
|---|---|---|---|---|---|---|---|---|---|
| Naïve Bayes | Dataset 1 | 3329 | 1235 | 1094 | 6943 | 82% | 85% | 86% | 86% |
| | Dataset 2 | 6562 | 7254 | 1063 | 21,751 | 77% | 75% | 95% | 84% |
| Logistic Regression | Dataset 1 | 3986 | 578 | 551 | 7486 | 91% | 93% | 93% | 93% |
| | Dataset 2 | 12,575 | 1241 | 1482 | 21,332 | 93% | 95% | 94% | 94% |
| K-NN | Dataset 1 | 3176 | 1388 | 3227 | 4810 | 63% | 78% | 60% | 68% |
| | Dataset 2 | 8043 | 5773 | 4203 | 18,611 | 73% | 76% | 82% | 79% |
| SVM | Dataset 1 | 3947 | 617 | 594 | 7443 | 90% | 92% | 93% | 92% |
| | Dataset 2 | 12,526 | 1290 | 1499 | 21,315 | 92% | 94% | 93% | 94% |

Therefore, analyzing the performance of Naïve Bayes on the two datasets, we observed that the accuracy was 82% for the first dataset and 77% for the second one. Therefore, for this specific algorithm, having more observations did not lead to a better accuracy, but on the contrary, it decreased its performance, proving that Naïve Bayes can perform on small-sized datasets as well. Moreover, for applying the same algorithm on the second dataset, we have obtained a better value for recall; however, all the other metrics are showing lower performance again.

In the case of Logistic Regression, the algorithm obtained an accuracy over 90% in both cases; however, the accuracy on the second dataset surpassed the results of the first one by 2pp. We can also see that F1-score is better for the second set of data.

For K-Nearest Neighbor, the accuracy obtained for the first dataset was 63%, while it was 73% for the second dataset. Even though we have observed an increase of 10pp when the number of observations was higher, none of the results are remarkable.

For the SVMs, the accuracies obtained for both datasets were over 90%, the first run resulting in an accuracy of 90% and the second one resulting in an accuracy of 92%. Training the algorithm on more observations led to an increase in accuracy of 2pp.

Comparing the performance of the algorithms on the two datasets, we can see that the second one managed to provide a better performance than the first for three out of

the four algorithms, which means that the best results can be obtained while more data are used, and also when this is collected from multiple sources, hence being more diverse. At the same time, it is well known that machine learning, especially supervised learning, performs better when the amount of data is higher.

Finally, in the short analysis above, we have seen all models from the perspective of Precisions, Recalls, F1 Scores and Balances Accuracies.

Precision calculates the ratio between the true positives and all the positives, more exactly [25]:

$$Precision = \frac{TP}{TP + FP}$$

where:

$TP$ is the number of true positive cases;

$FP$ is the number of false positive cases;

Recall is the measure of the model correctly identifying true positives. Therefore:

$$Recall = \frac{TP}{TP + FN}$$

where:

$TP$ is the number of true positive cases;

$FN$ is the number of false negative cases;

Accuracy will show as the ratio of the total number of correct predictions and the total number of predictions:

$$Accuracy = \frac{TP + TN}{TP + FP + TP + TN}$$

where:

$TP$ is the number of true positive cases;

$TN$ is the total number of predictions;

While we do know that accuracy is the most common performance metric used for classifications algorithms, this might not be the right metric when the classes are imbalanced. On other hand, the recall should be used when we know with certainty that we want to minimize the false negatives, while precision for minimizing the false positives. Finally, F1-score is also introduced, and this will use both precision and recall values, giving a balance between precision and recall [37].

Therefore, considering the results of these indicators presented above, we can conclude that, for our specific analysis, Logistic Regression and SVM were the most performant algorithms, obtaining very good results for the first and second dataset. For these two algorithms, both the accuracy and the F1-score were above 90% on the first dataset, while for the second dataset an increase of ~2pp was observed.

However, it might be important to mention the fact that SVM will take more time, so it will consume more resources to train and predict the same amount of data. In particular, for our scenario, SVM has taken more than four times more to perform the same steps as Logistic Regressions.

In addition to the initial analysis and tests performed with the algorithms chosen, we would like to also try to improve these, so that we can observe if even better results can be obtained.

In order to do this, we have also implemented the process of hyperparameter tuning. Since we are not immediately aware of the optimal model architecture that works for a given model, we could also explore a range of possibilities. The parameters that define the model architecture are generally named in machine learning hyperparameters.

Model training is actually a learning process in which a model can recognize patterns in the training data that has been provided, and also predict output for the new data based on these patterns. In addition to hyperparameter setting, model architecture directly impacts the execution time for training and testing a model. Therefore, due to their impact on the model performances, and since also the best set of values for parameters is unknown,

hyperparameter setting has become an important and challenging problem in machine learning algorithm implementation.

For our case, and also according to the new findings in recent literature, a sophisticated approach was observed, and that is to observe the optimal hyperparameter setting for the learning process and apply it to the model afterwards. This approach treats hyperparameter settings as a problem of optimizations, where the main objective is to maximize the machine learning performances as well as to minimize errors.

Moreover, many recent studies have proposed different techniques to find a set of optimal hyperparameters, and the most popular ones are grid search and random search. With GS (grid search), all the possible combinations of hyperparameter values are tested in order to find the best setting, based on the specified lower and upper bounds of each hyperparameter, along with a step size that will also be specified, which will form the hyperparameter space. A downside for this method is that the combination of the hyperparameters will increase multiplicatively as the number of hyperparameters increases. Therefore, this makes GS very time consuming, requiring a high computational cost. The alternative for this method is the RS (random search) method that randomly selects the values of the hyperparameter. An important drawback of this method would be the fact that this does not use the information from prior tests to select and decide on the next set, and it does not use a strategy either to predict the next trial [38].

Considering the details presented above, we have implemented in the following section both grid search and random search as hyperparameter tuning for the algorithms chosen for our text classification. Each model has been run again in order to observe the best parameters. After observing the best ones, they have also been implemented for training the model again. The last step of this process was to observe the newly obtained accuracy for both methods, for all our algorithms.

According to the above-mentioned steps followed for each of our algorithms, we can see the final results in Table 6.

**Table 6.** Summary of results after introducing hyperparameter tuning.

| Algorithm | Method | TN | FP | FN | TP | Best Accuracy |
|---|---|---|---|---|---|---|
| Naïve Bayes | Grid Search | 7009 | 6797 | 1308 | 21,516 | 77.75% |
| | Random Search | 12,216 | 1590 | 3209 | 19,615 | 77.67% |
| Logistic Regression | Grid Search | 12,820 | 986 | 1384 | 21,440 | 93.44% |
| | Random Search | 12,526 | 1277 | 1452 | 21,372 | 92.51% |
| K-NN | Grid Search | 8503 | 5303 | 4596 | 18,228 | 72.89% |
| | Random Search | 7712 | 6094 | 4896 | 17,928 | 72.43% |
| SVM | Gird Search | N/A | N/A | N/A | N/A | N/A |
| | Random Search | N/A | N/A | N/A | N/A | N/A |

Considering the above results, we would also like to mention that we tried to implement the same action plan for the SVM algorithm; however, even after around six hours (which is six times more than the time consumed by the most consuming algorithm before), the SVM with hyperparameter tuning did not provide any results. Therefore, even though this might have provided good results (considering that the initial training also provided a very good accuracy and F1 score), the resources consumed and the time to process are not optimal for our scenario, especially since Logistic Regression provided a very good performance and took around an hour for the time taken to train the same amount of data.

Last, but not least, we can conclude that, after trying to improve our models using this last step of hyperparameter tuning, we managed to obtain similar performance values to the initial ones obtained in the first step. Moreover, we can also observe the fact that grid search provided better results, even if the difference is very little in comparison to random search.

*3.6. Deployment*

Finally, if we accept these datasets as training datasets for our organization, we can say, based on the above performance indicators, that it is optimal to proceed on the prediction phase using the Logistic Regression. The best performance was obtained again for Logistic Regression even after applying the hyperparameter tuning (especially since SVM did not manage to provide any results on the training model performance).

Of course, once we implement this and manage to retrieve even more data from the constant reviews that are provided on the platforms mentioned, we can resume our process and monitor this algorithm's performance.

## 4. Discussion and Conclusions

As we have seen until now, this research highlights an automatic way to retrieve data and then applies different methods of analysis to provide insights for business audiences. In addition, it is critical to mention that this approach is not using pre-labeled data for the end-to-end process (and the lack of pre-labeled data is a well-known issue for current supervised algorithms in text and image processing), apart from one validation step that was possible due to the manual labeling for a subset of our input data. However, our main research is trying to consolidate a labeled dataset and use a mix of techniques. Some other research has been conducted around this kind of implementation as also explained by Borg et al. [39], also due to the fact that high amounts of public and veridic text data are very hard to be found with sentiment associated with the text. Moreover, even if the implementation is similar, we can observe in the current research a different combination of techniques.

Now, based on the above details, we conclude that the first part has focused on choosing the right social media platform with business and job-related information, retrieving the data in the most optimal and fast way, and also transforming the unstructured data into structured data, and loading it into a place from where it can be easily used for further analysis. Considering that the initial retrieval of the data was in an unstructured format that would have represented a downside for the next analysis we wanted to perform, we also introduced some steps to transform and aggregate the data, so that they can be used as wished for further analysis.

We have observed that another challenge for our research was to take data from all the concerned sites since not all of them provide the same structure of the data. We found an option to improve the web scraper in order to use parameters and retrieve data from different websites—with the mention that only common information such as title, feedback, and rating can be used in the analysis. Another problematic step is the process of retrieving the data constantly or in a real-time manner so that we can always have the most recent feedback from current or former employees. The process can be automated to have data in almost-real time by creating a schedule to run at a predefined time interval. For our study, the web scraping technique applied helped us to automatically retrieve data by creating an algorithm that can be used to extract data from different websites such as Indeed or Glassdoor for any company.

Nonetheless, the current dataset can be also improved by adding data from multiple websites (not only two sources as we have in the current analysis) or from multiple companies. Another approach can be trying to create a combined dataset with information from multiple sources using the common elements, such as the title of the review, feedback, and information regarding the author, location and date. Moreover, for future work, we are thinking of improving our study and overcoming the challenge of obtaining real data by integrating the current algorithm into an orchestration tool.

The second part of our research was mainly focused on using the data obtained and realizing sentiment analysis on it, so that we can understand the bigger picture of the opinions a company receives from current and former employees. However, considering the techniques existing in the area, we were forced for the moment to apply initially a lexicon-based approach, since we did not have a high amount of pre-labeled data with

variable sentiment added for each review that was provided, a technique that does not seem to have been used before with the same purpose. Nonetheless, we have also introduced a quick validation step for our initial method that verifies text block lexicon method performance. This step managed to conclude that our method has a performance similar to the other implementations based on the same approach. Therefore, we agreed that, based on this, we can continue to use this process to label all our data, even though we also agree that this performance can and should be improved. Therefore, the lexicon-based approach helped us add this variable with positive or negative sentiment for all the reviews and, in this way, we managed to overcome the challenges of not having a pre-labeled variable for sentiments that we needed for the following part. In addition, future research in the area might take into consideration the most recent challenges regarding the limited coverage of sentiment words for several domains [40,41]. Moreover, to improve this kind of analysis, we can include in our future work self-learning method, which becomes more popular in this field, since these types of unstructured data are very difficult to find with labels associated. Recent research associated with this domain has been developed just in the past years; therefore, we can also try to extend our study using this technique. Moreover, we can try to understand even more on similar research regarding methods to improve the lexicon-based method's performance, apart from the data pre-processing and preparation, which has already been followed.

Lastly, the next step in our process, respecting the previously presented methodology, was to continue with the newly created data as pre-labeled data (assuming that this new dataset was our base dataset) and apply different supervised machine learning techniques on them, such as Naïve Bayes, KNN, Logistic Regression and SVM. At this moment, trying on the new dataset, different algorithms also provided us the chance to compare them and observe that, in our scenario, Logistic Regressions and SVM provided the best results. However, the run time of the SVM was almost four times higher than the one for Logistic Regression. Hence, if computational resources are also a selection criterion, Logistic Regression, even though not a benchmark among supervised models in text mining research, would be a better recommendation considering the results achieved, comparable with the ones obtained for SVM, but at a fraction of the cost. Additionally, considering that the performance was higher in three out of the four analyzed cases for the second dataset, which had more observations, we can conclude that machine learning classification algorithms perform better when they are trained on a higher volume of data. Moreover, we also wanted to explore the possibility of improving the performance of these four algorithms; therefore, we have also implemented hyperparameter tuning so that we can understand if better results can be obtained for our models. After this step where we introduced two different methods for this technique, we observed that we managed to obtain slightly better performances, with Logistic Regression being again the best algorithm in terms of performance. Finally, SVM did not manage to provide any results in around six hours of running it with the same resources that the other algorithms managed to use to provide outputs in less than an hour. Therefore, we concluded that, for high amounts of data and also for the current scenario, this algorithm should not be considered anymore.

Additionally, to improve current research and results, other authors have proposed the parallelization and distribution of text classification. Moreover, other research in this field has also shown the fact that combining more classifiers for the analysis can improve classification accuracy [42].

Finally, we managed to obtain, as initially described, an end-to-end solution for our business need that can clearly determine an initial step for every organization that wants to take advantage of social platform data and use them in the context of big data analytics, without being dependent on other data collection and platforms available or different teams to initiate the analysis. Of course, this study can be extended by correlating the initial data we have obtained with internally collected text data from one organization, so that we can also have different purposes of the text provided.

Last, but not least, considering the artificial intelligence engaged in this entire process, we can also continue evolving the machine learning techniques used until now in this natural language processing research and classification problem. Moreover, we can extend the current use of artificial intelligence to introduce some pre-trained deep learning techniques and observe which can provide better performance. For the moment, the current research focuses on more classic approaches, as these perform very well in similar research, and the time and resources needed are far less than for other methods.

**Author Contributions:** Conceptualization, L.G.T. and A.V.; methodology, L.G.T. and C.A.V.; software, L.G.T. and A.V.; data curation, A.V.; writing—original draft preparation, L.G.T. and A.V.; writing—review and editing, A.R.B. and C.A.V.; visualization, L.G.T.; supervision, A.R.B. All authors have read and agreed to the published version of the manuscript.

**Funding:** This paper was co-financed by The Bucharest University of Economic Studies during the PhD program.

**Data Availability Statement:** All data used in the analysis were extracting from the following two websites: Indeed (https://www.indeed.com/cmp/Amazon.com/reviews) and Glassdoor (https://www.glassdoor.com/Reviews/Amazon-Reviews-E6036.htm) using a web scraping tool developed by the authors. All the information used for the analysis is available at the websites mentioned above and can be accessed by creating a free account.

**Conflicts of Interest:** The authors declare no conflict of interest.

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
