# Peer review of "Big Data ETL Process and Its Impact on Text Mining Analysis for Employees’ Reviews"

_applsci, doi:10.3390/app12157509_

Round 1

Reviewer 1 Report

I suggest including the results obtained from the experiments in the abstract.

modify figure 5, 6 and 7 since it is not of high quality.

Table 4 is not adequate since the rows are cut and not all the information is visible.

In general I think it is an interesting research but I suggest to modify the tables and images.

Author Response

Dear reviewer,

Thank you a lot for your feedback on our paper. We really appreciate your help in improving our work!

Firstly, regarding your suggestion on improving the abstract, we have added a brief presentation of the results inside it.

Moreover, we have modified the figures that were not correct or visible, and also corrected all the other mistakes regarding the figures and the spelling. 

Thank you once again for your time and we look forward to hearing from you!

Best regards

Reviewer 2 Report

Presented paper proposes the idea of using the sentiment analysis for the reviews obtained from the job related portals. In the first part of the work, they propose an architecture to obtain and transform the data from the sources and in the second part, the sentiment analysis in collected data is performed.

Most of my remarks are related to the analytical part. Usually, it is good to follow some methodology (e.g. CRISP-DM, or SEMMA, or other), to clearly define the business and analytical goals, to be able to properly evaluate the models - I'm missing this in the paper. 

Also mutiple details are missing in the particular analytical steps - e.g. how many records were used to label the sentiment, how they were selected. Preprocessing should be also explored more, as the lexicon based approaches are sensitive to preprocessing techniques. Also, I am missing any kind verification of the sentiment labeling - at least a part of the data could be labelled manually (or using a crowdsourcing) to properly evaluate the correctness of the sentiment labels. 

Then, the classifiers to predict the sentiment are not described well. How they were trained/validated/evaluated? Which input features were used and how (how was the text represented?), details about the preprocessing (as authors used methods that require different preprocessing, therefore differences in the performance). Were the models validated, hyperparameters tuned? Which metric is crucial for a given task and why - therefore which of the models fits the task the best?

The manuscript also suffers from multiple formal issues - e.g. tables formatting, low-res figures (e.g., Fig 5), figures out of the paper borders, source codes inserted as figures - in general, I would not use the source codes in the scientific paper at all - they could be rather provided publicly in some repository (e.g. github) and paper shoud focus more on a description, than a documentation. 

Author Response

Dear Mr./Mrs.,

Thank you a lot for your feedback on our paper. We really appreciate your help in improving our work!

First of all, we have understood that we omitted  in the beginning the methodology we followed for our analysis, therefore, we have added this to clarify the steps we have followed on pages 5 and 6 (from line 240 to line 280). In addition, we have also shared more details on the steps we followed on the two approaches and what part of the methodology was followed for it. 

Secondly, we have provided more details on the records used for each analysis, as well explaining more on the preprocessing steps that we have done for our data, fact that can be seen on page 13 (from line 406 to line 428), but also on the improvements we have done on the explanation for each of the lexicon analysis and machine learning methods (from line 405 to to 507, and also from 520 to 540).

In addition, we have understood that concerns regarding the missing verification as we don’t use any pre-labelled data, however, as we also expressed before, we have tried this approach of combining the lexicon method with machine learning algorithms to provide an end-to-end solution, without including any other data or manual work that could affect the entire optimization of the process proposed. 

Moreover, we have added more details and clarifications for the training settings and configurations and data split for training and testing, that can be seen on page 18 (from line 587 to line 590), as well as more information on the indicators we have chosen to test the performance of the algorithms and which of these are the most important for our scenario and why, fact expressed more on pages 20, 21 and 22 (from line 646 to line 685). 

Finally, we have also addressed the formatting issues that you have shared with us by removing the code we were sharing on the research, renamed the 2 tables as figures, we have also updated all the tables to use the same format, resized some figures and made new print screens for the histograms. 

Thank you once again for your time!

Best regards,

Reviewer 3 Report

In this work, the authors present an approach an end-to-end process in retrieving and analysing employee reviews using text mining and sentiment analysis.

At first it is described how to collect and transform the data proposed for analysis. The second part analyzed the data by using text mining techniques to retrieve business insights.

For the sentiment analysis, a lexicon approach is presented and 4 different machine learning algorithms are tested and the results compared.

Also the paper is all in all well written, its content is not that convincing for me.

The paper does not show any novel approaches but is more or less an extensive guideline how to deal with this kind of approaches. As the authors have, however put a lot of effort into a good description of the examined problem and discuss the pros and cons of different kinds of ways how to tackle the different tasks like information retrieving and the analysis of the data, the work might nevertheless be worth publishing.

In the following, I list the different points that have attracted my attention and should be improved.

I don't s any additional value by providing the code snippets, displayed in Figure 2, 4, 9. The code can not be reused by the readers as essential definitions and functions are missing. As it is crucial for these kind of papers to provide extensive insights into the source code, you should provide access to the code via some open-source platform like github.com or gitlab.com.

One of my main concerns, however is regarding "3.2.2 Machine learning techniques for sentiment classification".

The authors stated correctly in the previous parts of the paper that supervised machine learning algorithms can not be efficiently applied for sentiment analysis due to the lack of labels.

In this section, however they present 4 algorithms (Naive Bayes, Logistic Regression, KNN and Support Vector Machine) by using the datasets described during the paper as training data while the labels are created by the previously presented lexicon approach.

At first, the 4 algorithms are never explained. It would be nice to at least provide a rough introduction into the different kinds of algorithms and their differences. 

My second concern regarding this section is the used labeling. Yes, there is no other way of labeling the data as a manual labeling is too time consuming. But there is a high probability that the labels generated by the lexicon approach are not entirely correct which is influencing the performance of the machine learning algorithms (garbage in - garbage out). Without a clean dataset, it does not make sense to study the machine learning algorithms. As the lexicon approach is working well, I also don't see the benefit in switching to a more complex machine learning approach.

The 4 machine learning methods that are used in this paper more or less standard methods. In the last years however Deep NLP models have become the new state of the art (e.g. PEGASUS). In this paper they are only mentioned in the last section "4. Discussion and conclusions". Why did the authors not try one of this approaches? There are some models that don't need much training data as one can combine pre-trained models with few-shot learning or other techniques.

Other issues:

l. 335: 3.. Methods and results: Remove the second dot

l. 263, Figure 3.: You write "(left – Indeed.com; right – Glassdoor)". There is no left and right image in the figure. Please correct this.

l. 280: Can you describe the characteristics of a Databricks cluster in one or two sentences? This is not common knowledge.

l. 398, Figure 10: The captions of some of the plots in this figure are sometimes odd. E.g. "Average of Sentiment Polarity by Culture". The same goes for Pay Benefits and Work Balance, where in addition to the _ in Sentiment_Polarity, Pay Benefits and Work Balance should be written as two words.

l. 443: Sometimes you write "Naïve Bayes" and sometimes Naive Bayes. Please be consistent.

Author Response

Dear Mr./Mrs.,

Thank you a lot for your feedback on our paper. We really appreciate your help in improving our work!

First of all, regarding the code snippets and figures that were not correctly displayed, we have removed them from the paper and also placed the scripts on git for the interested ones. 

Secondly, we have understood the need to explain the algorithms that we have used for our analysis, therefore we have added a short description for each of them on page 17 (from line 559 to line 583). In addition, we have added more details on the preprocessing steps that we have followed and the way we tried to implement the algorithms in the best way for our purpose (starting with line 405, continuing on line 520).

Moreover, we have observed the concern regarding the fact that our final results might not be 100% correct, and, as we shared before, we have tried (with this approach of combining the lexicon method with machine learning algorithms) to provide an end-to-end solution, without including any other data or manual work that could affect the entire optimization of the proposed process. 

Last, but not least, on the Deep NLP proposal, we would like to share with you that for the moment, the current research focuses on more classic approaches as these perform very well in similar research as shared on page 24 (on line 775), and also the time and resources needed are way less than for other methods. However, we can extend the current use of artificial intelligence and introduce some pre-trained deep learning techniques and observe which can provide better performance. At the same time, we wanted to prove the effect of the ETL process on some well-known classification algorithms and propose an automated scraping solution for multiple sites. All these ideas were also added to the conclusion section, on page 24. 

Finally, we also considered and adjusted the writing issues, the figures that were not correct, the differences in writing „Naïve Bayes”, as well as providing more information on Azure Databricks on page 8 (from line 333 to line 337). 

Thank you once again for your time and we look forward to hearing from you!

Best regards

Round 2

Reviewer 1 Report

I consider the modifications to be appropriate

Author Response

Dear reviewer,

Thank you for your time on our research.

In order to improve our work, and also to upgrade it according to the suggestions and requirements received from the other reviewers, we have done the following changes:

  1. Regarding data mining methodology, we have introduced on line 247 a well-known methodology used in data mining (and also one of the ones you initially proposed), and in the chapter “Methods and results”, we have described every step of the methodology and how we benefit from it to correctly conduct our research and tests.
  2. For the initial check of sentiment labeling using lexicon-based approach, we have introduced a validation step of this method (from line 534 to line 558). We managed to do this by manually labeling a subset of our data and  applying the same method on this subset first, just to have the chance to compare the predicted values to the real ones. 
  3. Regarding the pre-processing part, we have both explained more of all the steps that were done in order to address the pre-processing part and provided justification based on similar research on why the algorithms we used were chosen and why we chose this kind of implementation for our end-to-end process.
  4. For the way in which we can handle the missing values for our scenario, we have added more clarifications from line 470 to line 477.
  5. Finally, regarding the model's optimizations, we also introduced two ways of realizing hyperparameters tuning, according to each of the algorithms. We can see this part in detail starting with line 866 and finishing on line 1021. Based on this additional step, we also managed to extract and add even more conclusions regarding the best algorithms to use for high amounts of data.

Thank you for your time!

Best regards,

Reviewer 2 Report

In the modified version of the paper, the authors added a few paragraphs of the text regarding to some of the previously mentioned remarks, raised during the first round of review. 

Unfortunately, I cannot say, that the added text actually addressed the raised questions. In general, during the previous review, I recommended to reject the paper. If the paper is still to be resubmitted, in such case not only a few paragraphs of new text are expected to be added, but mostly the new or re-designed experiments to be conducted, methodology adapted etc.. None of this I am seeing in the updated version of the manuscript. Also, added text in my opinion do not address the issues from the previous review. To sum it up:

- methodology - the authors still do not follow any standard analytical methodology (as proposed) and do not clearly define the business goal and corresponding analytical goal. Therefore, the actual usefullness of the trained models still cannot be verified. 

- no additional experiments were conducted to verify the sentiment labelling

- very little details about the preprocessing was added - mostly generic descriptions of particular steps, but what was important - the reasoning, why and how it was applied in this case and what was achieved

- classifiers description was added - but mostly generic text about the used algorithms. However, no reasoning why those were selected, no additional info regarding the hyperparameters, training, optimization and evaluation regarding to the business goal (as it is still missing) is not provided. 

- formatting of the paper and formal issues were improved, but multiple issues still persist (at least in my copy of the paper).

Author Response

Dear reviewer,

Thank you for the recent feedback provided on this research.

Please find below the improvements we have done to address the recent feedback received:

  1. Regarding data mining methodology, we have introduced on line 247 a well-known methodology used in data mining (and also one of the ones you initially proposed), and in the chapter “Methods and results”, we have described every step of the methodology and how we benefit from it to correctly conduct our research and tests.
  2. For the initial check of sentiment labeling using lexicon-based approach, we have introduced a validation step of this method (from line 534 to line 558). We managed to do this by manually labeling a subset of our data and  applying the same method on this subset first, just to have the chance to compare the predicted values to the real ones. 
  3. Regarding the pre-processing part, we have both explained more of all the steps that were done in order to address the pre-processing part and provided justification based on similar research on why the algorithms we used were chosen and why we chose this kind of implementation for our end-to-end process.
  4. Finally, regarding the model's optimizations, we also introduced two ways of realizing hyperparameters tuning, according to each of the algorithms. We can see this part in detail starting with line 866 and finishing on line 1021. Based on this additional step, we also managed to extract and add even more conclusions regarding the best algorithms to use for high amounts of data.

Thank you for your time!

Best regards,

Reviewer 3 Report

All in all the authors have improved their work. The important parts were addressed and it is now worth to be published. 

Nevertheless there are some small details that should be corrected. 

- Line 229: The URLs for the Indeed and Glassdor Web-Sites are missing here. They are only mentioned later in Table 1. It would be nice to have at least two foot notes with the Links to the web pages.

- Line 253: Checking if null values exist in the dataset. If yes, those can be replaced with the mean or the median of the variable.

What kind of Null Values? Missing reviews? How can they be replaced with a mean or median? It would be nice to have this explained in more detail.

- Line 258: thru -> through

Author Response

Dear reviewer,

Thank you for the recent feedback provided on this research.

Please find below the improvements we have done to address the recent feedback received:

  1. Regarding the missing URLs, please see them attached as footnotes for both sites.
  2. For the methodology used and applied, we have updated it and explained how this was applied and respected through the entire data mining process, a fact observed in chapter “Methods and results”.
  3. For the way in which we can handle the missing values for our scenario, we have added more clarifications from line 470 to line 477.

Thank you for your time!

Best regards,

Round 3

Reviewer 2 Report

In this version of the paper, most of the remarks from previous rounds were addressed.

Now I have just a few, mostly formal remarks:

- I would recommend to document the results of evaluation of all of the models in other format than screenshots from python output - it is common practice in scientific papers to provide e.g. tables with metrics (or for confusion matrixes), rather than direct screenshots from the pyhon output (which are, by the way, in low-quality resolution).

- also I would check the quality of the other figures as some of them (at least in my copy of the paper) appears to be in lower quality than others

Author Response

Dear Mr/Mrs.,

Thank you for the constructive feedback provided on this research.

Please find below the improvements we have done to address the recent feedback received:

  1. As you suggested, we removed the screenshots and presented the results as tables with metrics ( see Table 4 and Table 6 ).
  2. We tried to make some adjustments to the figures and tables. In some cases, for example the dashboard shown in Figure 10, the images are much larger (and interactive) in reality and it is difficult to clearly show all the details in the small space provided by the width of the text.

Best regards!